# Chlorogenic Acid Alleviated AFB1-Induced Hepatotoxicity by Regulating Mitochondrial Function, Activating Nrf2/HO-1, and Inhibiting Noncanonical NF-κB Signaling Pathway

**DOI:** 10.3390/antiox12122027

**Published:** 2023-11-22

**Authors:** Qianqian Wang, Tianxu Liu, Matthew Koci, Yanan Wang, Yutong Fu, Mingxin Ma, Qiugang Ma, Lihong Zhao

**Affiliations:** 1State Key Laboratory of Animal Nutrition and Feeding, Poultry Nutrition and Feed Technology Innovation Team, College of Animal Science and Technology, China Agricultural University, No. 2. West Road Yuanming Yuan, Beijing 100193, China; wqq@cau.edu.cn (Q.W.); liutianx@cau.edu.cn (T.L.); wyn@cau.edu.cn (Y.W.); yutong_fu@cau.edu.cn (Y.F.); mamingxin@cau.edu.cn (M.M.); maqiugang@cau.edu.cn (Q.M.); 2Prestage Department of Poultry Science, North Carolina State University, Raleigh, NC 27695, USA; mdkoci@ncsu.edu

**Keywords:** chlorogenic acid, aflatoxin B1, hepatotoxicity, antioxidant, anti-inflammatory, mitochondrial function

## Abstract

Aflatoxin B1 (AFB1), a kind of mycotoxin, imposes acute or chronic toxicity on humans and causes great public health concerns. Chlorogenic acid (CGA), a natural phenolic substance, shows a powerful antioxidant and anti-inflammatory effect. This study was conducted to investigate the effect and mechanism of CGA on alleviating cytotoxicity induced by AFB1 in L-02 cells. The results showed that CGA (160 μM) significantly recovered cell viability and cell membrane integrity in AFB1-treated (8 μM) cells. Furthermore, it was found that CGA reduced AFB1-induced oxidative injury by neutralizing reactive oxygen species (ROS) and activating the nuclear factor erythroid 2-related factor 2 (Nrf2)/heme oxygenase-1 (HO-1) signaling pathway. In addition, CGA showed anti-inflammatory effects as it suppressed the expression of inflammation-related genes (*IL-6*, *IL-8*, and *TNF-α*) and AFB1-induced noncanonical nuclear factor kappa-B (NF-κB) activation. Moreover, CGA mitigated AFB1-induced apoptosis by maintaining the mitochondrial membrane potential (MMP) and inhibiting mRNA expressions of *Caspase-3*, *Caspase-8*, *Bax*, and *Bax/Bcl-2.* These findings revealed a possible mechanism: CGA prevents AFB1-induced cytotoxicity by maintaining mitochondrial membrane potential, activating Nrf2/HO-1, and inhibiting the noncanonical NF-κB signaling pathway, which may provide a new direction for the application of CGA.

## 1. Introduction

Aflatoxins are a kind of metabolite produced by the fungi *Aspergillus parasiticus* and *A. flavus* [1]. Among the known aflatoxin derivatives (B1, B2, G1, and G2), aflatoxin B1 (AFB1) possesses the highest toxicity, and it was classified as a Group 1 (carcinogenic to humans) carcinogen by the International Agency for Research on Cancer (IARC) [2]. Several studies suggested that exposure to AFB1 causes liver oxidative injury by the excessive production of reactive oxygen species (ROS) in organs [3]. In general, low levels of ROS play an important role in maintaining physiological conditions, while high levels of ROS are associated with pathological conditions due to their damage to subcellular structures and biomolecules [4]. Similarly, ROS induced by AFB1 caused an imbalance in the antioxidant system and damaged biological molecules including lipids, proteins, and DNA in cellular systems [5]. Moreover, AFB1 induced mitochondrial damage as it could uncouple mitochondrial oxidative phosphorylation and increase the opening of mitochondrial permeability transition pores [6]. The combination of these negative effects induced cells to produce potentially catastrophic genetic alterations, eventually resulting in apoptosis or autophagy [7]. Therefore, it is important to explore effective methods for alleviating AFB1-induced damage to human health. 

Nutritional antioxidants, including polyphenols, antioxidant vitamins, and trace elements, were targeted for mitigating AFB1-induced toxin damage. For example, it was reported that a Se supplement in broilers’ diet alleviates AFB1-induced liver damage by increasing the activities of antioxidant enzymes like catalase (CAT), glutathione peroxidase (GSH-Px), superoxide dismutase (SOD), and the molecular regulation of the death receptors pathway [8]. Vitamin E was shown to ameliorate AFB1-induced nephrotoxicity in rats, probably by the correction of the resultant oxidative stress parameters as well as the inhibition of the protein expression of caspase-3 [9]. In addition, the growing body of studies has shown that polyphenols such as carotenoids and quercetin are able to mitigate AFB1-induced hepatic damage by mediating the expression of antioxidant enzymes [10,11]. However, it is noted that many existing studies focus on phenotypic measures, while the underlying molecular mechanisms related to nutritional antioxidants, including chlorogenic acid (CGA), are still poorly understood.

CGA, a major family of phenolic acids derived from the esterification of cinnamic acids, is abundant in coffee, fruit, and vegetables. It was found that CGA has various biological activities such as free radical scavenging and antioxidant, antibacterial, anti-inflammatory, and cardiovascular protection [12,13,14]. Previous studies reported that CGA could protect the liver from damage caused by chemicals or lipopolysaccharide [15]. Further research showed that CGA has a protective effect on the liver of rats by modulating inflammation and redox homeostasis [16]. In addition, researchers found that CGA protects against aluminum-induced cytotoxicity by enhancing the activities of antioxidant enzymes (SOD and CAT) and increasing the expression of nuclear factor erythroid 2-related factor 2 (Nrf2) in primary hippocampal neuronal cells [17]. Based on these results, CGA may be a potential preventive agent for alleviating AFB1-induced hepatotoxicity. However, the protective effect of CGA on the cytotoxicity induced by AFB1 and its underlying mechanism remain elusive. Therefore, this study aims to investigate the antioxidant effect and the underlying action mechanism of CGA on a hepatic cell exposed to AFB1.

## 2. Materials and Methods

### 2.1. Materials and Reagents 

CGA (purity, 98.0%) was purchased from Solarbio (Beijing, China). AFB1 was obtained from Pribolab (Singapore City, Singapore). Fetal bovine serum (FBS) and Roswell Park Memorial Institute (RPMI) 1640 medium were purchased from GIBCO (Grand Island, NE, USA). Antibodies were obtained from Proteintech (Wuhan, China).

### 2.2. Cell Culture

Human liver cells L-02 were purchased from Tongpai biotechnology co, ltd (Shanghai, China). Cells were maintained in RPMI 1640 containing 10% fetal bovine serum (FBS), 100 U/mL penicillin, and 100 μg/mL streptomycin (GIBCO, Grand Island, NE, USA) at 37 °C with 5% CO_2_. 

### 2.3. Cell Viability Assay

To select suitable AFB1 and CGA concentrations, the cell viability was determined with a CCK-8 assay. Briefly, cells (1 × 10^4^ cells per well) were seeded in 96-well plates for 24 h, and then cells were treated with different concentrations of AFB1 (from 1 to 32 μM) for 24 h or different concentrations of CGA (from 10 to 320 μM) for 6 h. For the CGA + AFB1 experiment, after being pre-incubated with culture medium (with or without CGA) for 6 h, the supernatant was removed, and then the cells were treated with culture medium (with or without AFB1) for another 24 h. Thereafter, the CCK-8 assay was performed by adding 110 μL of fresh medium containing 10 μL of CCK-8 solution to the cells and incubating them for 1 h at 37 °C. The absorbance was measured at 450 nm using a microplate reader (Thermo Labsystems, Pittsburgh, PA, USA). Cell viability was calculated as follows: Cell viability (100%) = (OD treatment/OD control) × 100%.

### 2.4. Determination of Lactate Dehydrogenase (LDH), Aspartate Aminotransferase (AST), and Alanine Aminotransferase (ALT) Activity

The L-02 cells were seeded in 6-well plates (5 × 10^5^ cells per well) for 24 h; after four treatments (Control, CGA, AFB1, CGA + AFB1), the supernatant was collected and centrifuged at 3000 r/min for 10 min. The activities of LDH, AST, and ALT were measured with assay kits (Nanjing Jiancheng Bioengineering Institute, Nanjing, China) according to manufacturer’s protocol.

### 2.5. RNA-Seq Analysis

Total RNA of cells was extracted with TRIzol reagent (Vazyme, Shanghai, China) according to the manufacturer’s instructions. Samples were submitted to Shanghai Majorbio Bio-pharm Technology (Shanghai, China) for mRNA purification, library preparation, and sequencing. The data were analyzed on the free online platform of Majorbio Cloud Platform (URL: www.majorbio.com, accessed on 20 June 2020).

### 2.6. Oxidative Stress Analysis

Cells were collected with 300 μL Dulbecco’s Phosphate-Buffered Saline (DPBS) and crushed 10 times continuously with ultrasonic cell crusher (60%) after treatment. Then, the cell homogenate centrifuged at 12,000× *g* for 10 min at 4 °C to collect the supernatant. Total protein concentrations of parallel samples were measured using a BCA Protein Assay kit (Beyotime, Shanghai, China). Malondialdehyde (MDA) level, total antioxidant capacity (T-AOC), and SOD activity were measured with assay kits according to the manufacturer’s protocol (Nanjing Jiancheng Bioengineering Institute, Nanjing, China). The production of cellular ROS was measured with a reactive oxygen species assay kit (Beyotime, Shanghai, China) according to the manufacturer’s instructions. Briefly, the treated cells were incubated with DCF-DA (10 mM) for 30 min at 37 °C. Then, cells were observed with a fluorescence microscope.

### 2.7. Mitochondrial Membrane Potential Determination

Cells were incubated with JC-1 solution for 20 min at 37 °C in the murk environment and were aspirated with the staining solution. The fluorescence intensity of cells in each well was tested by an inverted fluorescence microscope.

### 2.8. Quantitative Real-Time PCR

Total RNA of cells was extracted with TRIzol reagent (Vazyme, Shanghai, China) according to the manufacturer’s instructions. cDNA was synthesized using PrimeScript RT Master Mix (Vazyme, Shanghai, China). mRNA expression levels of genes were determined by quantitative real-time PCR (RT-qPCR) using SYBR Premix Ex Taq (Vazyme, Shanghai, China). Oligonucleotide primers of apoptosis-related genes (Caspase-3, Caspase-8, Caspase-9, Bax and Bcl-2) were designed based on databases of National Center for Biotechnology Information (NCBI) using Oligo (V 7.0) and synthesized by Sango Biotech Co., Ltd. (Shanghai, China). The relative mRNA expression levels of each target gene were calculated based on the expression of the housekeeping gene GAPDH using the 2^−ΔΔCt^ method [18]. Table 1 lists the primers of apoptosis-related genes used in this study.

### 2.9. Western Blotting

Total protein of cells was collected and determined using BCA protein detection kit (Beyotime, Shanghai, China) after treatment. Equal amounts of proteins were separated by sodium dodecyl sulfate polyacrylamide gel (gel concentration, 10%; intensity of electricity, 120 V; time, 60 min) electrophoresis (SDS-PAGE) and transferred onto the polyvinylidene difluoride (PVDF) membranes. The membranes were blocked with TBST buffer containing 5% skim milk powder for 1 h, followed by incubating with the primary antibodies (included anti-TRAF2 (concentration, 1:1000), anti-RelB (concentration, 1:3000), anti-p52 (concentration, 1:1000), anti-Nrf2 (concentration, 1:6000), anti-Keap-1 (concentration,1:5000), anti-HO-1 (concentration,1:3000), and anti-GAPDH (concentration, 1:8000)) at 4 °C overnight (Proteintech, Wuhan, China), and then further incubated with the secondary antibodies (Proteintech, Wuhan, China; concentration, 1:8000) for 1 h at room temperature. The immunoreactive bands were visualized by enhanced chemiluminescence and quantitatively analyzed by ImageJ2 software. 

### 2.10. Molecular Docking of CGA with the KELCH-like ECH-Associated Protein1 (Keap-1)-Nrf2 Complex

The structure data file (SDF) of CGA was downloaded from the PubChem website (Compound CID: 1794427) and converted to PDB format on OpenBabel 3.1.1. The structure of Keap-1-Nrf2 complex was downloaded from the RCSB Protein Data Bank (PDB ID: 2FLU). AutoDockTools1.5.7 software was used to modify the receptor protein, such as water removal and hydrogenation. Next, AutoDock Vina 1.1.2 was used to perform the molecular docking of the best ligand with each protein. Eventually, ligand binding flexibilities with the binding pocket residues were drawn using Discovery Studio 2019.

### 2.11. Statistical Analyses

All experiments were performed at least three times with similar results and the data obtained are expressed as the mean ± standard deviation (SD) unless otherwise specified. Analyses were performed using GraphPad Prism Version 7 software (GraphPad, San Diego, CA, USA). Statistical differences between control and the other groups were detected by one-way analysis of variance (ANOVA) followed by multiple comparisons. Differences were considered statistically significant when the *p*-values were ≤0.05.

## 3. Results

### 3.1. CGA Attenuated AFB1-Induced L-02 Cell Cytotoxicity

We first investigated the cytotoxic effect of AFB1 on L-02 cells. After a 24 h incubation, the viability of hepatocytes was significantly decreased with 8, 16, and 32 μM AFB1 treatment compared to the cells without AFB1 treatment. (Figure 1A). Therefore, 8 μM AFB1 was used in the following experiments. The cell viability in L-02 cells with the increasing concentrations of CGA is shown in Figure 1B. The cell viability after 6 h incubation was minorly variational when the concentration of CGA was lower than 160 μM. However, it was hazardous to L-02 cells when the concentration of CGA reached 320 μM. Consequently, 160 μM CGA was used in the following treatments. To investigate whether CGA could protect L-02 cells from AFB1-induced toxicity, the cells were pretreated with CGA for 6 h, and then exposed to AFB1 for 24 h. Remarkably, 160 μM CGA completely abolished the decrease in cell viability induced by 8 μM AFB1 (Figure 1C). Pretreatment of cells with CGA also reduced LDH release as well as intracellular levels of ALT compared with only AFB1-treated cells, indicating CGA could help maintain the cell membrane integrity (Figure 1D,E). These data demonstrate that CGA ameliorates the survival and function of L-02 cells with AFB1 exposure.

### 3.2. RNA-Seq Analysis

To investigate the protective mechanisms of CGA on AFB1-induced L-02 cells, the RNA-seq was used to determine the whole gene expression pattern of L-02 cells. After AFB1 was treated, a total of 1716 genes were changed, including 1172 upregulated genes and 544 downregulated genes compared to the control group (Figure 2A,C). There were 28 upregulated and 117 downregulated genes in the CGA + AFB1 group compared with the AFB1 group (Figure 2B,C). However, there were minor differences in genes between the control group and CGA group at a fold change of 1.8 per gene (*p* ≤ 0.05). According to the GO enrichment analysis (Figure 2D,E), the DEGs were classified into three main categories: biological process, cellular component, and molecular function. The most obvious difference was found in the biological process category and the mainly changed GO terms included “intrinsic apoptotic signaling pathway in response to DNA damage by p53 class mediator”, “apoptotic signaling pathway”, “molecular function regulator”, and “inflammatory response”. Among these changes, we summarized the functions of the main changes as “oxidative stress”, “inflammatory response”, and “apoptosis” (Figure 2F). 

### 3.3. CGA Alleviated AFB1-Induced Oxidative Damage by Activating Nrf2/Heme Oxygenase-1 (HO-1) Signaling Pathway in L-02 Cells

To investigate the effect of CGA on the oxidative stress reaction to cells induced by AFB1, the ROS levels of L-02 cells were measured in using a fluorescent probe. Imaging indicated that the percentage of DCF positive cells increased after AFB1 treatment (Figure 3A). Interestingly, the AFB1-induced enhancement in ROS content was remarkedly inhibited by CGA pretreatment. The high production of ROS is usually accompanied by a decrease in enzyme activity and membrane fluidity [19]. In the present study, cells in AFB1 group exhibited lower activities of T-AOC and SOD as well as higher concentrations of MDA (*p* ≤ 0.05) compared with control cells. However, pretreatment with CGA in AFB1-treated L-02 cells markedly promoted the activities of T-AOC and SOD but decreased the level of MDA compared with AFB1-treated L-02 cells (*p* ≤ 0.05) (Figure 3B–D). Thus, these results indicate that CGA could alleviate AFB1-induced oxidative damage to L-02 cells. Increasing evidence has indicated that the Nrf2/HO-1 signaling pathway plays a pivotal role in the protection of oxidative stress [20]. In order to explore the protective mechanisms of CGA on AFB1-induced oxidative damage to L-02 cells, a computational molecular docking analysis was performed in this work to evaluate whether there is any affinity between CGA and the Keap-1-Nrf2 complex. The result showed that there were high-affinity (9.4 kcal/mol) hydrogen binding events between the residues of ARG415, VAL512, VAL418, VAL465, ILE559, VAL606, and GLY367 in the Keap-1-Nrf2 complex and CGA (Figure 3G), which meant that CGA interacted with the Keap-1-Nrf2 complex. Moreover, the levels of Nrf2 and HO-1 were detected using Western blot analysis to further confirm the effects of CGA in the Nrf2/HO-1 signaling pathway. The results revealed that pretreatment with CGA significantly increased the protein expression levels of HO-1 and nuclear Nrf2 in L-02 cells compared with AFB1 treatment, without effects on the expression levels of Keap-1 and total Nrf2 (Figure 3H–M). Therefore, it indicated that CGA probably alleviated AFB1-induced oxidative damage by interacting with the Keap-1-Nrf2 complex, promoting nuclear translocation of Nrf2 and upregulating the expression of HO-1.

### 3.4. CGA Alleviated AFB1-Induced Inflammatory Response by Preventing Noncanonical Nuclear Factor Kappa-B (NF-κB) Pathway in L-02 Cells

Inflammatory response is an important mechanism in the toxic effects of AFB1. In this research, clustering analysis showed that AFB1 notably elevated expressions of inflammation-related genes in L-02 cells, which were significantly suppressed by CGA pretreatment. In addition, noncanonical NF-κB signaling pathway-related genes including *TRAF2*, *RelB*, and *p52* were significantly increased in the AFB1 group compared with those in the control group. However, these changes were substantially recovered by CGA pretreatment (Figure 4A). The gene set enrichment analysis (GSEA) also supported that AFB1 exposure significantly activated the NF-κB complex, while it was inhibited by pretreatment with CGA (Figure 4B,C). RT-qPCR results revealed that AFB1 exposure caused up-regulations on the mRNA expressions of *IL-6*, *IL-8*, and *TNF-α* in L-02 cells, whereas the mRNA abundances of these genes were down-regulated in AFB1-treated cells with CGA pretreatment (*p* ≤ 0.05) (Figure 4D–F). KEGG pathway enrichment analysis identified the pathway connected with “NF-κB signaling pathway” as well (Figure 4G,H). Based on this, we speculated that CGA may regulate AFB1-induced inflammatory response via the noncanonical NF-κB signaling pathway. Therefore, the expression levels of TRAF2, RelB, and p52 were detected using Western blot analysis to verify the bioinformatics analysis results. The results revealed that the expression levels of TRAF2, RelB, and p52 were significantly increased in the AFB1 group, but these changes were effectively alleviated in AFB1-treated cells with CGA pretreatment (Figure 4I–L). Collectively, these results suggested that CGA could prevent noncanonical NF-κB pathway activation induced by AFB1 in L-02 cells.

### 3.5. CGA Alleviated AFB1-Induced Apoptosis in L-02 Cells 

Alteration of mitochondrial membrane potential (MMP) is an important link in oxidative stress-induced apoptosis [21]. When MMP is higher than the cell membrane spot, the probe aggregates at the cell membrane and emits red fluorescence. However, the probe excites green fluorescence when the membrane potential drops. We observed that green fluorescence in AFB1-treated cells was increased, suggesting that AFB1 treatment triggered mitochondrial damage. After pretreatment with CGA, the fluorescence of AFB1-treated cells was nearly recovered to the normal group (Figure 5A). In addition, apoptosis is usually associated with proapoptotic and antiapoptotic proteins of the Bcl-2 family, initiator caspases (Caspase-8, -9, and -10), and effector caspases (Caspase-3, -6, and -7) [22]. In our study, L-02 cells treated with AFB1 increased the mRNA expressions of *Caspase-3*, *Caspase-8*, *Bax*, and *Bax/Bcl-2* compared with control cells. Interestingly, pretreatment with CGA in AFB1-treated L-02 cells decreased the mRNA expressions of *Caspase-3*, *Caspase-8*, *Bax*, and *Bax/Bcl-2* compared with the AFB1-treated cells (Figure 5B–G). These data indicated that CGA could alleviate AFB1-induced apoptosis by maintaining MMP and inhibiting the expressions of apoptosis-related genes in L-02 cells. 

## 4. Discussion

AFB1, a kind of mycotoxin produced by fungi, is always detected from grains such as groundnuts and maize without suitable storage [23]. This toxic metabolism is recognized as the most hazardous because it imposes acute or chronic toxicities on humans and thus causes great public health concerns [24,25]. It has been reported that AFB1 has a variety of adverse health effects such as immunotoxicity, neurotoxicity, and hepatotoxicity, causing childhood stunting, liver and renal injury, immunosuppression, and even cancer [26,27]. Several studies have shown that AFB1-exo-8,9-epoxide (AFBO), a metabolite of AFB1 in the liver, could induce mutations in the oncogene *p53*, which is associated with cell cycle, DNA repair, apoptosis, and autophagy in human hepatocytes [28,29]. Hence, how to alleviate AFB1-induced hepatocyte injury remains to be further studied. CGA, a kind of dietary polyphenol, shows powerful antioxidant and anti-inflammatory effects on animals’ bodies [30]. Moreover, numerous studies have confirmed that CGA could protect cells against oxidative stress by activating the cellular antioxidant defense system and counterbalancing the production of ROS. However, the protective mechanism of CGA on AFB1-induced cytotoxicity has not yet been clearly elucidated. Therefore, the effect and related mechanism of CGA on AFB1-induced hepatic cytotoxicity were explored in this study. We found that CGA addition could alleviate AFB1-induced cell damage with the increase in cell viability and the decrease in LDH release, confirming the protective effect of this polyphenol on cell membrane integrity. This is in accordance with a previous study which revealed that CGA addition could alleviate cytotoxicity of DON in IPEC-J2 cells with a decrease in LDH release [31]. 

To further explore the protective mechanism of CGA treatment on the AFB1-treated L-02 cells, RNA-seq and bioinformatics analyses were carried out in this work. GO analysis showed that CGA alleviated AFB1-induced cell damage by regulating oxidative stress, apoptosis, and inflammatory response. The imbalance between oxidation and anti-oxidation is one of the landmark events in the development of AFB1-induced hepatotoxicity. Oxidative stress is mainly manifested as excessive production of ROS and a decrease in the concentrations of antioxidant enzymes and an increase in the concentration of MDA [32]. Several studies have revealed that the mitigation of oxidative stress is conducive to curing damaged liver cells and a variety of therapeutic agents have significant antioxidant effects [33,34]. Cho et al. (2017) suggested that CGA ameliorated the incidence of photo-aging in UV radiation–exposed fibroblasts through decreasing the ROS production in mitochondria [35]. CGA has also been shown to attenuate LPS-induced oxidative stress through increasing the activities of GSH and SOD, and the GSH/GSSG ratio [36]. Accordingly, CGA could alleviate AFB1-induced oxidative injury in L-02 cells by directly neutralizing ROS production and indirectly modulating the activities of the antioxidant enzymes. 

Nrf2 is the key regulator of intracellular ROS and proinflammatory cytokine expression [37]. In the physiological state, Nrf2 is ubiquitinated through binding with Keap-1 in the cytoplasm. In the presence of oxidative stresses, Nrf2 then translocates into the nucleus and forms a coactivator complex to promote transcription and mRNA expressions of antioxidant and detoxifying genes, including *HO-1*, *GSH*, *NQO1*, and *GCLC*. Under homeostatic conditions, *HO-1* expression is low or absent in most cells. However, it is highly upregulated by most cells in response to a vast number of pro-oxidant stimuli and provides protection against oxidative damage [38]. A previous study reported that CGA reduced radiation-induced apoptosis and DNA damage via Nrf2 activation in hepatocellular carcinoma [39]. CGA has also been shown to prevent diabetic nephropathy by suppressing oxidative stress and inflammation through activation of the Nrf2/HO-1 signaling pathway and inhibition of the NF-κB signaling pathway [40]. Additionally, CGA has been found to prevent acute liver injury via the upregulation of Nrf2 and the inhibition of NLRP3 inflammasome activation [41]. In the present study, it was found that there were high-affinity (9.4 kcal/mol) hydrogen binding events between the Keap-1-Nrf2 complex and CGA. CGA interacting with Keap-1 might lead to the release of Nrf2 from the Keap-1-Nrf2 complex, which is essential for Nrf2 activation. In addition, the protein levels of Nrf2 in nucleus and its corresponding downstream HO-1 were significantly increased after CGA treatment, suggesting that the protective role of CGA in AFB1-induced oxidative injury may be in part related to its activation of the Nrf2/HO-1 pathway through facilitating Nrf2 nuclear translocation and up-regulating the expression of HO-1.

Inflammation is always considered to interact with oxidative stress. Inflammation increases oxidative damage, which in turn aggravates inflammation [42]. Numerous studies have indicated that over-expressions of cytokines such as IL-1, IL-6, and TNF-α may promote liver inflammation during chronic liver injury [43]. In this study, AFB1 significantly activated the relative mRNA abundances of *IL-6*, *IL-8*, and *TNF-α* in L-02 cells, which was in agreement with the previous study [44]. However, CGA addition was able to remit cell inflammation caused by AFB1 exposure, proving CGA possess anti-inflammatory functions. Considerable evidence suggested that CGA can regulate the inflammation caused by some factors. In the model of mouse inflammatory bowel diseases (IBD), CGA showed a dose-dependent decrease in expression and nuclear translocation of the NF-κB subunit, which was accompanied by the suppression of systemic inflammation in mice with colitis [45]. CGA was also reported to have a protective effect on myocardial infarction by reducing the inflammatory response and exerting antioxidant activity [46]. Another study indicated that CGA exerted protective effects against hypoxanthine and potassium oxonate-induced hyperuricemia in mice, which was attributed to its ability to suppress NF-κB activation [47]. In this study, GSEA profiling showed that NF-κB signatures were remarkably suppressed by CGA treatment. KEGG pathway enrichment analysis also identified that the related pathways connected with the “NF-κB signaling pathway”. Precise regulation of NF-κB signaling is crucial for regulating the expressions of many inflammatory factors and it is activated through the classical and noncanonical pathways [48]. Therein, the non-canonical NF-κB signaling cascade relies on the inducible processing of the NF-κB2 precursor protein (p100) and mediates the activation of the p52/RelB NF-κB complex [49]. Moreover, during the inflammatory response, the activation of NF-κB signaling pathways can induce the release of a variety of pro-inflammatory cytokines [50]. In the present study, the expression levels of TRAF2, RelB, and p52 were significantly increased in the AFB1 group compared with those in the normal group, while CGA supplementation reversed those changes. Collectively, these results suggested that CGA could alleviate AFB1-induced inflammatory response in L-02 cells by inhibiting the noncanonical NF-κB signaling pathway. 

It has been shown that mitochondria are the primary site to produce ROS and excessive ROS accumulation could disrupt the MMP, leading to mitochondrial dysfunction, thus triggering apoptosis [51]. Polyphenol has been, in many contexts, shown to directly affect mitochondrial activity and maintain MMP [52]. A previous study reported that CGA pretreatment improved H_2_O_2_^●−^ induced intestinal mitochondrial damage, increased ROS production, and decreased cytochrome C release [53]. In line with this outcome, our study found that CGA can improve AFB1-induced hepatic mitochondrial damage and protect the integrity of mitochondrion effectively. Moreover, it has been reported that mitochondria-dependent apoptosis is usually associated with caspase activation [54]. Increased mRNA expressions of *Caspase-3*, *Caspase-8*, and *Bax* have been observed in response to AFB1 treatment, which is consistent with a previous study suggesting that AFB1 exposure induced ROS-dependent caspase-mediated apoptosis in normal human cells [55]. Interestingly, the pretreatment of CGA markedly decreased the mRNA expressions of *Caspase-3*, *Caspase-8*, *Bax*, and *Bax/Bcl-2* in AFB1-treated cells. Consequently, these results suggested that CGA might alleviate AFB1-induced apoptosis by maintaining MMP of cells and inhibiting the expressions of apoptosis-related genes (Figure 6).

## 5. Conclusions

This study demonstrated that AFB1 exposure induced oxidative stress, inflammatory response, and apoptosis, leading to a decrease in cell viability eventually. Moreover, CGA prevented AFB1-induced oxidative damage by neutralizing ROS production and modulating anti-oxidative enzyme activity, as well as activating the Nrf2/HO-1 signaling pathway. CGA alleviated AFB1-induced inflammatory response by inhibiting the noncanonical NF-κB signaling pathway and CGA alleviated AFB1-induced apoptosis by maintaining MMP and inhibiting the expressions of apoptosis-related genes, ultimately restoring cell viability. 

## Figures and Tables

**Figure 1 antioxidants-12-02027-f001:**
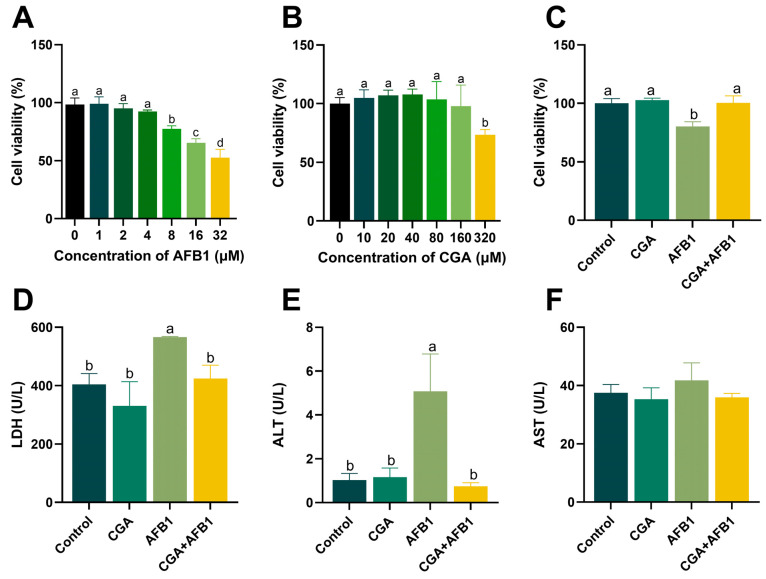
CGA attenuated AFB1-induced L-02 cell cytotoxicity. (**A**) the cell viability of L-02 cells after different concentrations of AFB1 incubations for 24 h, *n* = 6. (**B**) the cell viability of L-02 cells after different concentrations of CGA incubations for 6 h, *n* = 6. (**C**) effect of CGA on AFB1-induced cells viability in L-02 cells, *n* = 6. (**D**) the LDH concentration of L-02 cells in different treatments. (**E**) the ALT concentration of L-02 cells in different treatments. (**F**) the AST concentration of L-02 cells in different treatments. Note: No shoulder marker or same shoulder marker in column means the difference is not significant (*p* > 0.05), different shoulder markers mean the difference between groups was significant (*p* ≤ 0.05).

**Figure 2 antioxidants-12-02027-f002:**
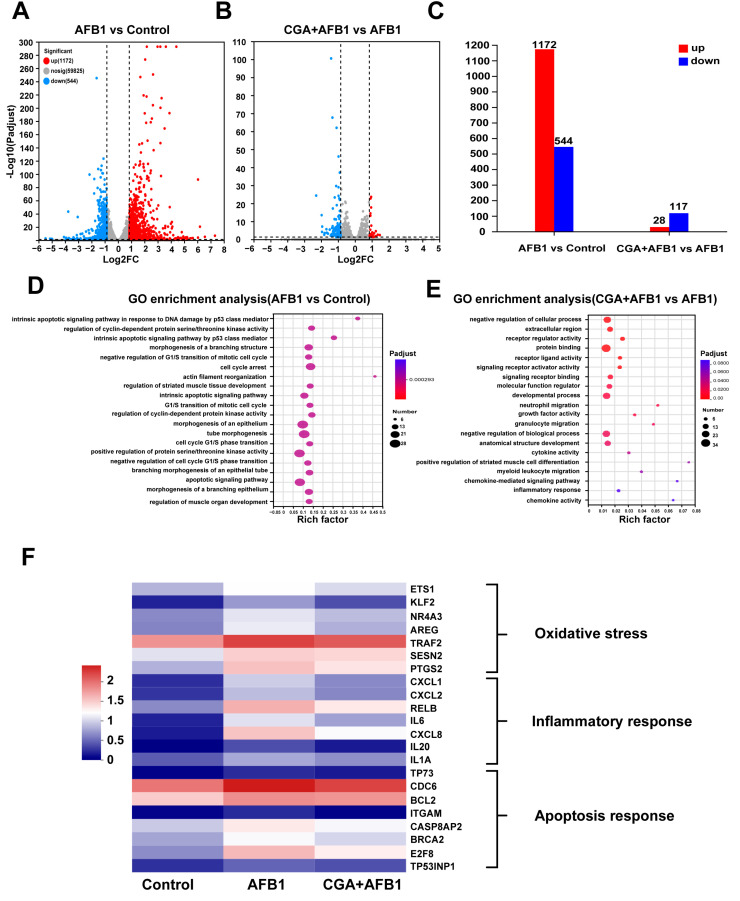
The RNA-seq of L-02 cells was used to determine the whole gene expression pattern. (**A**) the volcano plot shows the distribution of genes and the results of significant differences in genes in AFB1 group vs. control group. (**B**) the volcano plot shows the distribution of genes and the results of significant differences in genes in CGA + AFB1 group vs. AFB1 group. (**C**) the genes of up-regulation and down-regulation in this study, *p* ≤ 0.05. (**D**) the GO enrichment analysis of AFB1 group vs. control group. (**E**) the GO enrichment analysis of CGA + AFB1 group vs. AFB1 group. (**F**) heat map of the representative function.

**Figure 3 antioxidants-12-02027-f003:**
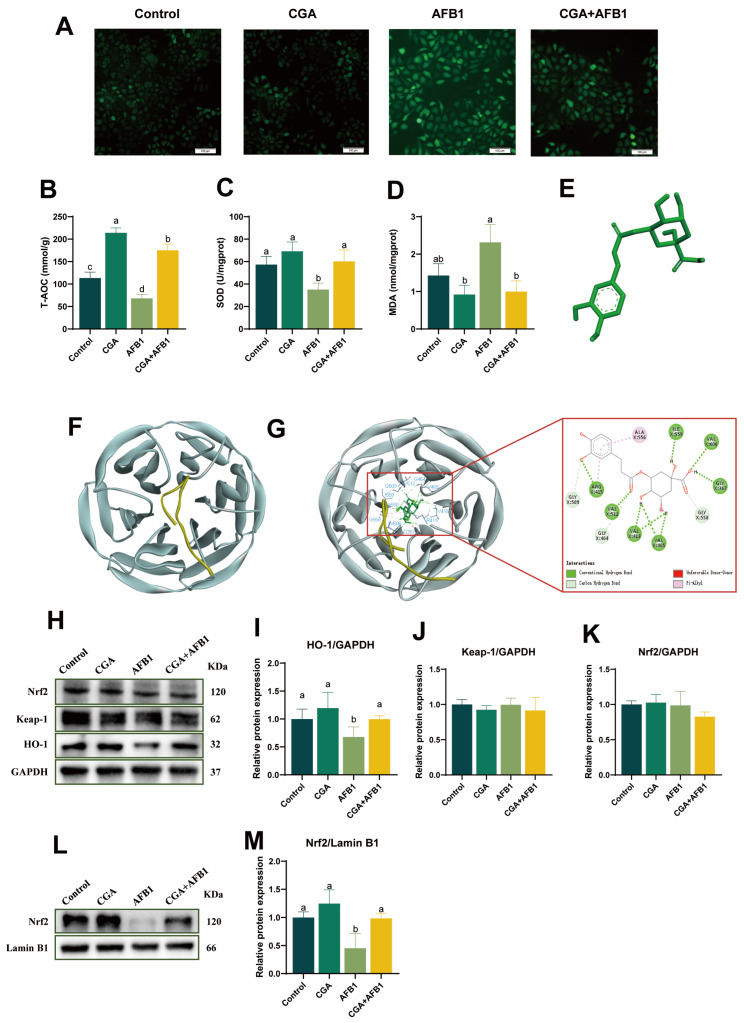
CGA alleviated AFB1-induced oxidative damage by activating Nrf2/HO-1 signaling pathway in L-02 cells. (**A**) effects of CGA on the ROS production of AFB1-induced L-02 cells. (**B**) the T-AOC activity of L-02 cells in different treatments. (**C**) the SOD activity of L-02 cells in different treatments. (**D**) the MDA concentration of L-02 cells in different treatments. (**E**) the modular structure of CGA. (**F**) the modular structure of Keap-1-Nrf2 complex. (**G**) the high interaction energy (−9.4 kcal/mol) between CGA and Keap-1-Nrf2 complex. (**H**–**K**) Western blots and quantification of total Nrf2, Keap-1, and HO-1. (**L**,**M**) Western blots and quantification of Nrf2 in the nuclear. Note: No shoulder marker or same shoulder marker in the same row means the difference is not significant (*p* > 0.05), different shoulder markers mean the difference between groups was significant (*p* ≤ 0.05).

**Figure 4 antioxidants-12-02027-f004:**
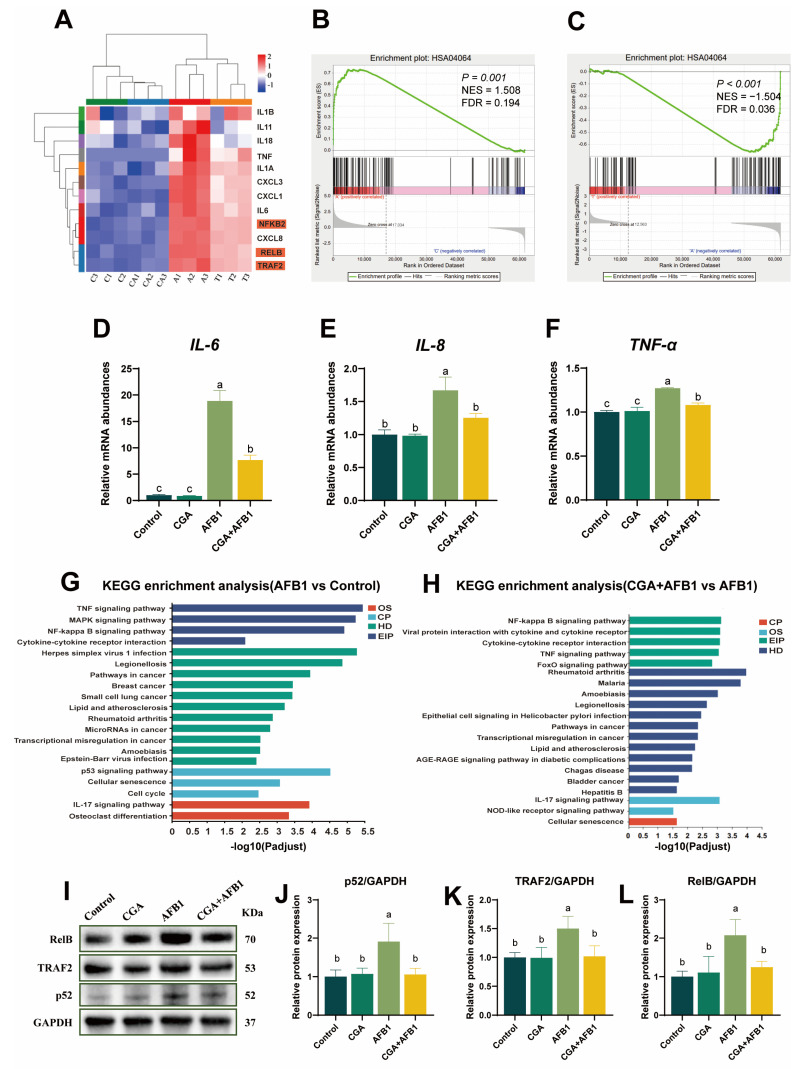
CGA prevented NF-κB pathway activation induced by AFB1 in L-02 cells. (**A**) clustering analysis of inflammation-related genes. (**B**) GSEA plot of a significant gene set associated with NF-κB-mediated signaling (Control vs. AFB1). FDR, false discovery rate; NES, normalized enrichment score. (**C**) GSEA plot of a significant gene set associated with NF-κB-mediated signaling (AFB1 vs. CGA + AFB1). (**D**–**F**) relative mRNA expressions of *IL-6*, *IL-8*, and *TNF-α* in L-02 cells. (**G**) KEGG pathway (AFB1 vs. Control) enrichment. “NF-κB signaling pathway” is shown with shadow red. (**H**) KEGG pathway (CGA + AFB1 vs. AFB1) enrichment. “NF-κB signaling pathway” is shown with shadow red. (**I**–**L**) Western blots and quantification of RelB, TRAF2, and p52. Note: No shoulder marker or same shoulder marker in the same row means the difference is not significant (*p* > 0.05), different shoulder markers mean the difference between groups was significant (*p* ≤ 0.05).

**Figure 5 antioxidants-12-02027-f005:**
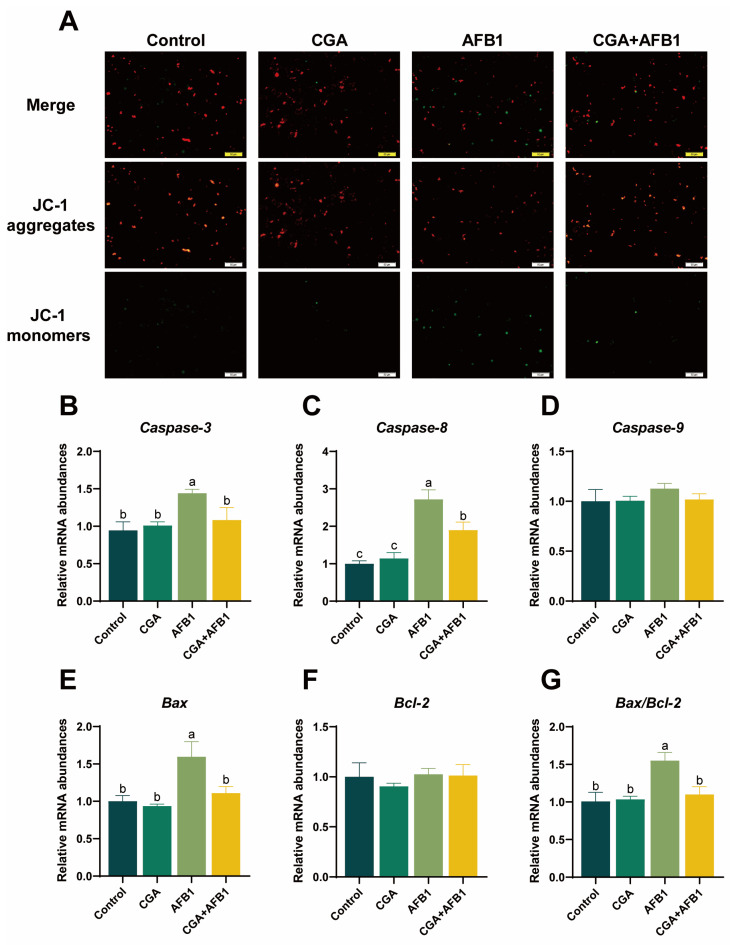
CGA alleviated AFB1-induced apoptosis in L-02 cells. (**A**) effects of CGA pretreatment on the mitochondrial membrane potential in AFB1-induced L-02 cells. (**B**–**G**) relative mRNA expressions of *Caspase-3*, *Caspase-8*, *Caspase-9*, *Bax*, *Bcl-2*, and *Bax/Bcl-2* in L-02 cells. Note: No shoulder marker or same shoulder marker in the same row means the difference is not significant (*p* > 0.05), different shoulder markers mean the difference between groups was significant (*p* ≤ 0.05).

**Figure 6 antioxidants-12-02027-f006:**
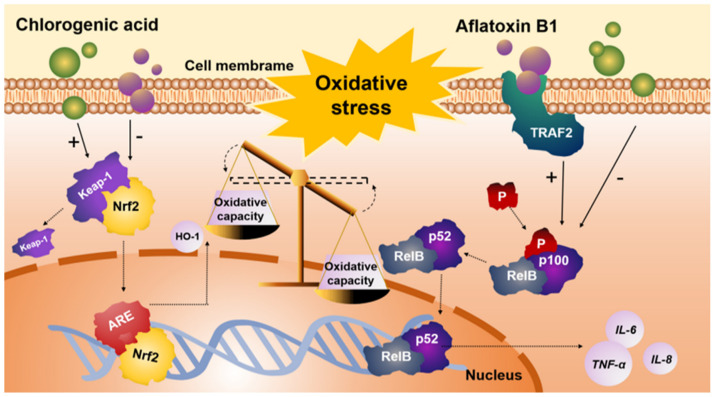
Schematic diagram of the protective effects of CGA on AFB1-induced cytotoxicity in L-02 cells.

**Table 1 antioxidants-12-02027-t001:** The primers of apoptosis-related genes used in this study.

Gene	Forward Primer (5′-3′)	Reverse Primer (5′-3′)	Gene Accession Number
*Caspase-3*	CCAAAGATCATACATGGAAGCG	CTGAATGTTTCCCTGAGGTTTG	XM_054350958.1
*Caspase-8*	CAAACTTCACAGCATTAGGGAC	ATGTTACTGTGGTCCATGAGTT	NM_033355.4
*Caspase-9*	TCCAGGAAGGTTTGAGGACC	CCCTTTCACCGAAACAGCAT	XM_011542273.4
*Bax*	CCCGAGAGGTCTTTTTCCGAG	CCAGCCCATGATGGTTCTGAT	XM_047439168.1
*Bcl-2*	GACTTCGCCGAGATGTCCAG	GAACTCAAAGAAGGCCACAATC	XM_054318967.1
*GAPDH*	CTCTGCTCCTCCTGTTCGAC	TTAAAAGCAGCCCTGGTGAC	NM_001357943.2

## Data Availability

Data are contained within the article. The transcriptomic data have been deposited into the NCBI sequence Read (SRA) with the dataset identifier SRP470990.

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
