# Peer review of "Chlorogenic Acid Alleviated AFB1-Induced Hepatotoxicity by Regulating Mitochondrial Function, Activating Nrf2/HO-1, and Inhibiting Noncanonical NF-κB Signaling Pathway"

_antioxidants, 2023, doi:10.3390/antiox12122027_

Round 1

Reviewer 1 Report

Comments and Suggestions for Authors

I would like to congratulate the authors for presenting their work very effectively.

Authors tried to address the molecular mechanisms associated with ROS breakdown. In this manuscript authors presented both gene and protein expression data defending their claims. 

Authors primarily addressed their objectives and data is original. Relevance of this work is not very high but considerable to the field.

This manuscript provides the basic idea of molecular pathways associated with aflatoxin induced inflammation and help others to focus at various choke points to reduce the inflammation.

Experimental design, results and discussion are very well written. Being said that I have following suggestions. 

1. In figures, specially in graphs, statistics are bit confusing. Please present them in standard format for the benefit of readers. 

2. Title of the manuscript claim to shown mitochondrial functional assays! and I haven't seen any functional assays measuring mitochondrial functions like mito-stress or measuring mitochondrial protein expression, etc. 

inclusion of any one functional assay justify the title. 

Author Response

Dear Editor and Reviewers,

We greatly appreciate the your careful work. Your comments have greatly improved the manuscript. We have highlighted the changes in the updated manuscript with a yellow background and the responses were listed below.

Comments  1: In figures, specially in graphs, statistics are bit confusing. Please present them in standard format for the benefit of readers.

Response 1: Thanks for reviewer’s comment. We have present the Figure 3 and Figure 4 in standard format.

Comments  2: Title of the manuscript claim to shown mitochondrial functional assays! and I haven't seen any functional assays measuring mitochondrial functions like mito-stress or measuring mitochondrial protein expression, etc. inclusion of any one functional assay justify the title. 

Response 2: Thanks for reviewer’s comment. It was reported that mitochondrial morphology and function are closely related, and an increase in mitochondrial membrane potential is consistent with improved mitochondrial function. therefore, monitoring mitochondrial membrane potential is a commonly method for assessing the functional status of mitochondria (Gao G, Wang Z, Lu L, et al. Morphological analysis of mitochondria for evaluating the toxicity of alpha-synuclein in transgenic mice and isolated preparations by atomic force microscopy. Biomed Pharmacother, 2017, 96: 1380-1388.). In this work, the mitochondrial membrane potential was determined by JC-1 solution to assess the mitochondrial function, so it justify the title.

Reviewer 2 Report

Comments and Suggestions for Authors

Dear Dr. Zhao,

The manuscript is a laborious study, including moderns and appropriate methods. I would add some punctual remarks, I hope you will find constructive.

The abstract should be reconsidered, row 29 repeats the information presented in row 13. In fact, the abstract is repeated twice. It should briefly describe some of the methods… and the result should be better organized.

Row 154. Please provide more data about WB. For example, the gel concentration, time and intensity of electricity used for electrophoresis and blot. Please make sure that the detail makes the protocol reproductible. Please include data about the vendor. Please provide the concentration of antibodies etc. 

Row 175 Statistics, please make clear when T test and when ANOVA have been used, what posttest?

In my opinion including a figure in the conclusion is unusual, it better fits into the discussions chapter  

Comments on the Quality of English Language

 Overall, I suggest a consistent improvement in English grammar and style at the introduction chapter and, partially, in results. In discussions and conclusions, I have no major complains related to English, but some phrases are just way to long, and difficult to follow. Please replace them by more concise sentences, easier comprehensible.

Author Response

Dear Editor and Reviewers,

We greatly appreciate the your careful work. Your comments have greatly improved the manuscript. We have highlighted the changes in the updated manuscript with a yellow background and the responses were listed below.

Comments 1: The abstract should be reconsidered, row 29 repeats the information presented in row 13. In fact, the abstract is repeated twice. It should briefly describe some of the methods… and the result should be better organized.

Response 1: Thanks for your careful checks. Based on your comments, we have delete the repeated abstract. The sentences that “RNA-seq analysis found that CGA regulated oxidative, inflammatory response and apoptosis in cells” has been deleted to simplify the methods. In addition, results were reorganized in the abstract.

Comments 2: Row 154. Please provide more data about WB. For example, the gel concentration, time and intensity of electricity used for electrophoresis and blot. Please make sure that the detail makes the protocol reproductible. Please include data about the vendor. Please provide the concentration of antibodies etc.

Response 2: Thanks to the reviewers’ opinion, we have provided more data about WB. “Total protein of cells was collected and determined using BCA protein detection kit (Beyotime, Shanghai, China) after treatment. Equal amounts of proteins were separated by sodium dodecyl sulfate polyacrylamide gel (gel concentration, 10%; intensity of electricity, 120 V; time, 60 min) electrophoresis (SDS-PAGE) and transferred onto the polyvinylidene difluoride (PVDF) membranes. The membranes were blocked with TBST buffer containing 5% skim milk powder for 1 h, followed by incubating with the primary antibodies (included anti-TRAF2(concentration, 1:1000), anti-RelB (concentration, 1:3000), anti-p52 (concentration, 1:1000), anti-Nrf2 (concentration, 1:6000), anti-Keap-1 (concentration,1:5000), anti-HO-1 (concentration,1:3000) and anti-GAPDH (concentration, 1:8000) at 4 ℃ overnight (Proteintech, Wuhan, China), and then further incubated with the secondary antibodies (Proteintech, Wuhan, China; concentration, 1:8000) for 1 h at room temperature. “

Comments 3: Row 175 Statistics, please make clear when T test and when ANOVA have been used, what posttest?

Response 3: Thanks for your careful checks. Based on your comments, we have corrected sentences as Statistical differences between control and the other groups were detected by one-way analysis of variance (ANOVA) followed by multiple comparisons.”

Comments 4: In my opinion including a figure in the conclusion is unusual, it better fits into the discussions chapter. 

Response 4: Thanks for reviewer’s comment. We have removed the figure 6 into the discussions chapter.

Comments 5: Overall, I suggest a consistent improvement in English grammar and style at the introduction chapter and, partially, in results. In discussions and conclusions, I have no major complains related to English, but some phrases are just way to long, and difficult to follow. Please replace them by more concise sentences, easier comprehensible.

Response 5: We revised the language and manuscript with the assistance of a native English spoken person.